# An Active Radar Interferometer Utilizing a Heterodyne Principle-Based Target Modulator

**DOI:** 10.3390/s25061711

**Published:** 2025-03-10

**Authors:** Simon Müller, Andreas R. Diewald, Georg Fischer

**Affiliations:** 1Department of Engineering, Trier University of Applied Sciences, 54293 Trier, Germany; s.mueller@et.hochschule-trier.de; 2Department of Electrical-Electronic-Communication Engineering, Friedrich Alexander Universität Erlangen, 91058 Erlangen, Germany; georg.fischer@fau.de

**Keywords:** cooperative radar, distance measurement, phase coherence, active radar target, reflective clutter suppression, sewer pipe monitoring

## Abstract

The Active Radar Interferometer (AcRaIn) represents a novel approach in secondary radar technology, aimed at environments with high reflective clutter, such as pipes and tunnels. This study introduces a compact design minimizing peripheral components and leveraging commercial semiconductor technologies operating in the 24 GHz ISM band. A heterodyne principle was adopted to enhance unambiguity and phase coherence without requiring synchronization or separate communication channels. Experimental validation involved free-space and pipe measurements, demonstrating functionality over distances up to 150 m. The radar system effectively reduced interference and achieved high precision in both straight and bent pipe scenarios, with deviations below 1.25% compared to manual measurements. By processing signals at intermediate frequencies, advantages such as improved efficiency, isolation, and system flexibility were achieved. Notably, the integration of amplitude modulation suppressed passive clutter, enabling clearer signal differentiation. Key challenges identified include optimizing signal processing and addressing logarithmic signal attenuation for better precision. These findings underscore AcRaIn’s potential for pipeline monitoring and similar applications.

## 1. Introduction

### 1.1. Terminology

The term secondary radar system is not used consistently worldwide. The Institute of Electrical and Electronics Engineers (IEEE) provides a definition in [1] and restricts the term in the American context to radar systems in which an interrogator sends out an encoded signal and requests a response. The cooperative radar target then responds with an encoded message. Examples of this include civilian air traffic-control systems or military friend-or-foe identification systems. In the European context, the term is used much more broadly. It encompasses all radar systems that do not rely solely on passive echoes to identify radar targets. Various technologies are employed in this context, enabling the echo at the radar target to be directly modified in a specific way or supplemented with additional information.

### 1.2. Brief History

Radar target technologies have evolved over the years and can be categorized into passive, semi-passive, and active devices, each tailored to specific operational needs. We give a short description of examples of each type in the following overview.

**Passive radar targets** reflect or scatter incident radar signals and are often able to add information by affecting the system’s passive characterization parameters such as modifying its reflection coefficient. This is the basic principle used, for example, in the Van Atta array [2] as one of the most simple principles or in the targets presented by Thornton and Edwards in 1998 and 2000 [3,4] by switching between different feeding lines of an antenna to modify its reflection coefficient.

**Semi-passive targets** add basic functionality, such as simple modulation schemes like phase-shifting signals by means of active components, but with minimal power requirements as there is typically no power added to the reflected signal. This type of backscatterer can be considered a passive component from an RF perspective. In 1977, Koelle and Depp [5] described the use of a cooperative target consisting of an amplitude-modulated backscatterer in combination with a Doppler radar. An advantage identified is that the modulation frequency does not need to be known a priori, nor does it require special stability, as the authors evaluate only phase changes in the baseband of the CW radar used. A half-wave dipole antenna is proposed as the backscatterer. If a pin diode is inserted as a load impedance in the middle and driven by a modulation frequency, the reflected RF signal is modulated by varying the reflection coefficient with this modulation.

**Active targets** go further by generating or significantly modifying signals, by adding energy to the echo. A variety of realizations can be found in Kossel et al. [6], Cantú [7], Fusco et al. [8], and Dadash [9,10], amongst other similar concepts.

Some fundamental principles are simplified in Figure 1. The two different one-port devices in Figure 1b can be implemented as transmission line segments, short circuits, or even oscillators.

The Switched Injection-Locked Oscillator (SILO) introduced by Vossiek and Gulden [11] has been used in several variants but can only be assumed coherent by approximation and only for a short time of several oscillation cycles.

A third group is synchronized and Tachymeter units. All nodes are constructed in a very similar or identical manner, as most of the necessary operations typically need to be performed on all units, requiring significant synchronization effort as demonstrated in [12,13,14], besides a number of variations of these. The measurement result is finally determined after merging data from each node. In addition to the radar itself, a communication channel is often required; alternatively, data can be transmitted using the radar signal as a carrier.

### 1.3. Preceding Development

In [15], the authors presented the basic principle of an active Radar Interferometer that consists of a radar base unit mounted at a fixed position that reads the signature of one or more radar targets. The units form a cooperative radar system designed for environments with high reflective clutter, such as pipes and tunnels.

This radar response system modifies the echo transmitted back to the base in a way that both unambiguity and phase coherence are given at any time during measurements. It eliminates the need for synchronization as the echo can be treated in almost the same way as a conventional passive echo. A separate communication is neither required nor intended.

The system leverages commercial semiconductor components, operating in the 24 GHz ISM band, and is designed to achieve a range of up to 120 m using an FMCW radar source.

Key advancements include a prototype that integrates amplitude modulation to suppress passive clutter and improve signal separation from the environment. Four data-acquisition techniques are discussed, including direct sampling, undersampling, secondary downconversion, and self-multiplication, each with trade-offs in cost, complexity, and data resolution. Experimental results demonstrated the radar’s precision in measuring pipe lengths, with deviations of less than 1.25% compared to manual tape measurements. The radar performed well in both straight and curved pipe scenarios, even under challenging conditions like moisture and obstructions. The capability to measure in bent pipes highlights its adaptability and the guiding effect of pipe walls on electromagnetic waves.

### 1.4. Innovation

In this article, the authors propose a topology that is improved in comparison with the preceding responder version. The development process revolves around the transition from the previously employed homodyne approach to a heterodyne operating principle. This modification allows the received signal to be processed and adjusted at an intermediate frequency before being transmitted back. By leveraging the heterodyne principle, the system will benefit from improved control over signal manipulation. The technological shift to signal manipulation in an intermediate frequency range offers several advantages:**Simplified Signal Processing:** Operating at a lower frequency range reduces complexity in signal treatment, enabling easier filtering, amplification, and amplitude control.**Enhanced Efficiency:** Reduced signal losses during processing lead to improved overall energy efficiency of the system.**Improved Isolation:** The separation between input and output signals is enhanced, minimizing interference and enabling cleaner signal transmission.**Greater Component Variety and Flexibility:** The use of intermediate frequencies broadens the range of compatible components, providing more flexibility in system design and implementation.**Cost and Size Reduction:** Lower frequency components tend to be smaller and less expensive, contributing to a more compact and cost-effective solution. This is a major issue in environments of limited space availability, such as sewer pipes.**Extensibility to Advanced Modulation Techniques:** This approach facilitates the integration of single-sideband modulators, which can further refine the system’s performance by reducing spectral bandwidth and improving signal clarity.

This strategic advancement in topology will lead to a focus on simplified signal processing, achieving better system efficiency, and expanding the radar’s application potential in challenging environments.

With the focus on the presentation of the modular hardware platform based on the heterodyne approach, two test series in free space and inside sewer pipes are elaborated on, analyzed, and discussed. These measurements to characterize the target performance therefore extend beyond the test of the prototype presented in [15].

## 2. Materials and Methods

This section outlines the materials and methods used in this work, detailing the system’s design and the underlying concepts. The following subsections describe the block diagram, a summary of key mathematical definitions from our preceding publication, system requirements, and the flexibility of the approach in terms of modification and measurement. Finally, the chosen topology is presented in detail.

### 2.1. Block Diagram and Mathematical Description

A high-level block diagram is provided to give an overview of the system architecture (Figure 2) and the signal flow at the responder (Figure 3). The mathematical description of the system has been previously detailed in a preceding publication [15], to which the reader is referred here to gain a better understanding of the background. Key definitions and principles relevant to this work will be briefly repeated for completeness. The variables used and their meanings are explained in Table 1.

The radar system targeting at the responder basically transmits a signal $y_{1}\left(t\right)$, which is described in the time domain as follows:(1)$$y_{1}\left(t\right)=A\cdot \cos\left(\omega_{\mathrm{HF}}\left(t\right)\cdot t+\varphi_{\mathrm{HF}}\right)$$

Considering path losses and the time of flight, the responder receives(2)$$y_{2}\left(t\right)=B\cdot \cos\left(\omega_{\mathrm{HF}}\left(t\right)\cdot \left(t-T_{\mathrm{oF}}\right)+\varphi_{\mathrm{HF}}\right)$$

In a first step, the received signal is downconverted to an intermediate frequency $f_{\mathrm{IF}}\left(t\right)=f_{\mathrm{HF}}\left(t\right)-f_{\mathrm{LO}}$. It is then amplitude-modulated with the known envelope frequency $f_{\mathrm{AM}}$, amplified, and finally upconverted again using the same local oscillator frequency $f_{\mathrm{LO}}$.

The goal is to achieve improved carrier suppression of the originally transmitted radar system frequency $f_{\mathrm{HF}}$.

Furthermore, the equation already known from the homodyne approach also applies here. By mixing with the oscillator signal and subsequent low-pass filtering, $y_{\mathrm{IF}}\left(t\right)$ is determined.(3)$$y_{\mathrm{IF}}\left(t\right)=\frac{B}{2}\cdot \cos\left(\omega_{\mathrm{HF}}\left(t\right)\cdot \left(t-T_{\mathrm{oF}}\right)-\omega_{\mathrm{LO}}\cdot t+\left(\varphi_{\mathrm{HF}}-\varphi_{\mathrm{LO}}\right)\right)$$

If no further amplification is applied, the modulation with the envelope now follows:(4)$$\begin{array}{cc} y_{\mathrm{IF}}^{\left[\mathrm{AM}\right]}\left(t\right)=\frac{B}{4}\cdot \cos(\omega_{\mathrm{HF}}\left(t\right)\cdot \left(t-T_{\mathrm{oF}}\right)+\left(-\omega_{\mathrm{LO}}+\omega_{\mathrm{AM}}\right)\cdot t & \ldots \\ \ldots +\left(\varphi_{\mathrm{HF}}-\varphi_{\mathrm{LO}}+\varphi_{\mathrm{AM}}\right)) & \ldots \\ \ldots +\frac{B}{4}\cdot \cos(\omega_{\mathrm{HF}}\left(t\right)\cdot \left(t-T_{\mathrm{oF}}\right)+\left(-\omega_{\mathrm{LO}}-\omega_{\mathrm{AM}}\right)\cdot t & \ldots \\ \ldots +\left(\varphi_{\mathrm{HF}}-\varphi_{\mathrm{LO}}-\varphi_{\mathrm{AM}}\right)) \end{array}$$

Next, $y_{\mathrm{IF}}^{\left[\mathrm{AM}\right]}\left(t\right)$ is upconverted again using the same local oscillator frequency. However, this leads to a further spreading into four superimposed oscillations. Since the oscillator’s phase at the upconverter does not necessarily match the phase at the downconverter, but both phases have a constant difference, this is accounted for as $\varphi_{\mathrm{LO}}^{\prime }$. This situation arises when the path lengths from the common signal source to the two mixers are not exactly equal.

The signal transmitted back to the receiver path of the radar sensor at its input is given by(5)$$y_{4}\left(t\right)=D\cdot \cos\left(\omega_{\mathrm{HF}}\left(t\right)\cdot \left(t-2\cdot T_{\mathrm{oF}}\right)+\varphi_{\mathrm{HF}}\right)$$
which is downconverted by multiplication with the signal $y_{1}\left(t\right)$. The classical radar parameters, namely range information and Doppler shift, can be determined through differential analysis of the two reference points, which are symmetrically positioned on either side of the identification oscillator’s frequency. The oscillation is described by a left-hand $u_{l}\left(t\right)$ and a right-hand $u_{u}\left(t\right)$ term: (6)$$\begin{array}{c} u_{l}\left(t\right)=\frac{A\;D}{4}\cos\left(\omega_{\mathrm{AM}}\cdot t+2\omega_{\mathrm{HF}}\left(t\right)\cdot T_{\mathrm{oF}}-\omega_{\mathrm{AM}}\cdot T_{\mathrm{oF}}+\varphi_{\mathrm{AM}}\right) \end{array}$$(7)$$\begin{array}{c} u_{u}\left(t\right)=\frac{A\;D}{4}\cos\left(\omega_{\mathrm{AM}}\cdot t-2\omega_{\mathrm{HF}}\left(t\right)\cdot T_{\mathrm{oF}}-\omega_{\mathrm{AM}}\cdot T_{\mathrm{oF}}+\varphi_{\mathrm{AM}}\right) \end{array}$$

Let $\omega_{\mathrm{HF}}\left(t\right)$ represent a linear frequency ramp with duration $\Delta T_{c}$ and bandwidth $BW=f\left(t_{0}+\Delta T_{c}\right)-f\left(t_{0}\right)$. The range *R* is then calculated as:(8)$$R=\frac{c_{0}\cdot \Delta T_{c}}{4\cdot BW}\cdot \left(f_{u}-f_{l}\right)$$

Here, $f_{l}$ and $f_{u}$ are the two frequency points associated with the oscillations $u_{l}\left(t\right)$ and $u_{u}\left(t\right)$.

The relationship in Equation (Equation 8) is largely identical to the classical radar solution but includes a factor of 4 instead of 2 in the denominator. This results from the fully differential consideration of the two frequency points — the left-hand frequency point corresponds, in principle, to the negative reference point in a classical radar, which is now shifted into the positive frequency domain by $f_{\mathrm{AM}}$.

To minimize interference, bandpass filtering is recommended since additional signal components may be expected alongside the desired reference points. Signal acquisition can then proceed, and several different approaches will be examined.

To determine the Doppler information, consider the summands in the oscillation terms of $u_{l}\left(t\right)$ and $u_{u}\left(t\right)$ that depend on $T_{\mathrm{oF}}$. Any change in the target’s range affects the signal propagation time $T_{\mathrm{oF}}$, while minimal movements, such as those on the order of the wavelength of $\omega_{\mathrm{HF}}\left(t\right)$, will not significantly shift the reference points $f_{l}$ and $f_{u}$. The two Equations (Equation 6) and (7) are extended to include the propagation time dependence τ (Doppler effect):(9)$$\begin{array}{cc} T_{\mathrm{oF}}\to & T_{\mathrm{oF}}\left(\tau \right) \end{array}$$(10)$$\begin{array}{cc} u_{l}\left(\tau \right)= & \frac{A\;D}{4}\cos\left(\omega_{\mathrm{AM}}\cdot t+2\omega_{\mathrm{HF}}\left(t\right)\cdot T_{\mathrm{oF}}\left(\tau \right)-\omega_{\mathrm{AM}}\cdot T_{\mathrm{oF}}\left(\tau \right)+\varphi_{\mathrm{AM}}\right) \end{array}$$(11)$$\begin{array}{cc} u_{u}\left(\tau \right)= & \frac{A\;D}{4}\cos\left(\omega_{\mathrm{AM}}\cdot t-2\omega_{\mathrm{HF}}\left(t\right)\cdot T_{\mathrm{oF}}\left(\tau \right)-\omega_{\mathrm{AM}}\cdot T_{\mathrm{oF}}\left(\tau \right)+\varphi_{\mathrm{AM}}\right) \end{array}$$

The first and fourth summands are constant in τ, while the third summand affects both terms (10) and (11) equally. Differential analysis leads to the desired relationship for the Doppler effect in the form of phase modulation ΔΦ(τ):(12)$$\begin{array}{cc} \Delta \Phi (\tau ) & =\Phi_{l}\left(\tau \right)-\Phi_{r}\left(\tau \right)\\ \; & =4\cdot \omega_{\mathrm{HF}}\left(t\right)\cdot T_{\mathrm{oF}}\left(\tau \right) \end{array}$$

The conclusion is that relative analysis of the two reference points ensures coherence. This allows both range and Doppler information to be determined without any loss in quality or precision.

### 2.2. System Requirements

The system requirements are outlined, including functional, technical, and environmental considerations essential for achieving the desired performance. From a circuit-design perspective, it is necessary to balance functionality, effort, and complexity. Initially, the requirements for the circuit are defined. This leads to the derivation of a block diagram and the functional segmentation of the components. Iterative minor adjustments may need to be made as required. It is advisable to first select integrated high-frequency circuits and mixers, as the available selection of usable components in this area is typically the most limited. Once the allowable signal levels and the necessary filter characteristics at the inputs and outputs of the central functional blocks are known, appropriate amplifier modules can be selected to complete the signal chain, and the calculation and design of filter circuits can be conducted. The circuit is completed with the necessary peripherals and power supply. Following this, the printed circuit board (PCB) is designed, considering optimal component placement and trace routing.

The requirement catalog is divided into three sections. The *General Requirements* section covers the system level. Subordinate requirements apply, respectively, to the radar base station and the active radar target.

#### 2.2.1. General Requirements

*1.01* The measurement system determines the distance to one or more active targets.*1.02* The range is at least 100 m.*1.03* The deviation of the measured distance from the actual distance shall be within ±5cm.*1.04* An ISM band is used.*1.05* The center frequency of the usable frequency range is 24.125 GHz, with a bandwidth of 250 MHz.*1.06* The signal levels and waveforms must be designed to ensure that the transmit power does not exceed 20 dBm EIRP.*1.07* The distance information of the active targets is transmitted in a frequency range that does not overlap with the frequency range of passive targets within the range of up to 100 m.*1.08* The measurement system consists of a PC with an application for data acquisition, a base station as a unit, and one or more active targets as individual units.*1.09* Communication between the PC application and the base station is established via a LAN connection.*1.10* Measurements are initiated from the PC application. The measurement value is computed in the PC software using the baseband time-domain signal.*1.11* The base station generates a suitable radar signal and processes the received signal in the baseband so that it can be digitally evaluated. Additional requirements are defined separately.*1.12* An active target modifies a received radar signal so that it can be clearly distinguished from passive reflectors within the range. Additional requirements are defined separately.

#### 2.2.2. Radar Base Station Requirements

*2.01* There is one transmit channel.*2.02* There are at least two receive channels.*2.03* The evaluation is based on the data from at least one receive channel.*2.04* The antennas are interchangeable. Therefore, all RF ports must be equipped with SMA connectors.*2.05* All antennas radiate in the same direction.*2.06* The transmit power is adjustable.*2.07* Power is supplied by a laboratory power supply, with a voltage range of 4.5 V to 5.5 V.*2.08* The design is modular. The following modules are implemented: antenna array (1), RF module (2), ADC module (3), controller module (4), digital interface (5). Modules (1)–(2) and (3)–(5) can each be implemented as combined units.*2.09* Module connections are made using plug-in or coaxial (SMA) connectors.*2.10* Modules (1) and (2) are designed so that they can be mounted at the end of a pipe with an inner diameter of 150 mm.

#### 2.2.3. Active Radar Target Requirements

*3.01* The active target is designed using the heterodyne method and employs a double-sideband modulation scheme.*3.02* There is one receive channel and one transmit channel.*3.03* All antennas radiate in the same direction.*3.04* The transmit power is adjustable.*3.05* Power is supplied by a laboratory power supply or a battery (voltage range 10 V to 16 V). The input must include protection against undervoltage, overvoltage, and reverse polarity.*3.06* The design is modular. The following modules are implemented: antenna array (1), RF module (2), reconfigurable IF module (3), power supply (4).*3.07* Module connections are made using plug-in or coaxial (SMA) connectors.*3.08* The IF module (3) consists of a carrier PCB with at least three slots. The central slot is occupied by a mixer module including an AM oscillator, and the remaining slots can optionally be equipped with a filter, amplifier, attenuator, or a combination of these.*3.09* Modules (1) and (2) are designed so that they can be mounted at the end of a pipe with an inner diameter of 150 mm.

### 2.3. Presentation of the Chosen Topology

The chosen topology is presented, highlighting the rationale behind its selection and the advantages it offers for this specific application.

#### 2.3.1. Software Flowchart

The flow diagram is illustrated in Figure 4. The base station acts as the server, while the application serves as the client. Measurement initiation is triggered by the PC application. After successfully establishing a connection to the base station (specifically, the microcontroller), hardware configuration and calibration are initiated. Once completed, the application can trigger a new measurement. Following the request, it waits for the complete transmission of raw data. To maintain the required timing intervals, the entire process up to data transfer is controlled by the microcontroller.

Starting with a linearly increasing ramp, the local oscillator frequency is set to the lowest frequency $f_{\mathrm{start}}$ within the available band. The duration of each ramp step is controlled by a timer, which specifies the constant length of each step using an interrupt routine. At the end of each timer interval, a new analog-to-digital conversion is triggered. Immediately afterward, the local oscillator is adjusted to the frequency of the next ramp step (frequency increment Δf). The cycle restarts with the timer overflow. This sequence (conversion, frequency step, time synchronization) ensures a constant interval, with the conversion timing shifted to the end of each step to account for oscillator settling time. The voltage value captured by the ADC is stored in the controller’s RAM. Once the ramp’s upper frequency $f_{\mathrm{stop}}$ is reached, the described process is interrupted, and the transmitter is turned off. All data points are transferred to the application, forming the baseband time-domain signal. Based on this, the desired distance information is calculated by the application, and the measurement result is displayed. Subsequently, a new measurement can be initiated, making execution in a constant time interval reasonable.

#### 2.3.2. Radar Frontend

A standard radar frontend should be used if possible. Our implementation is given in Figure 5. Therefore, when specifying the parameters for amplitude modulation on the responder side, its frequency $f_{\mathrm{AM}}$ must be located within the passband of the receiver’s mixer stage.

This consists of a PLL-controlled local oscillator, which feeds both the TX antenna and the mixer of the downconverter in the receiver path as another essential component. The antenna feeds for the TX and RX antennas are routed to SMA connectors to allow for testing various antenna configurations.

The amplitude modulation by the responder, as described, must also be taken into account in the analog-to-digital conversion. Four applicable methods for this are described in [15]. In addition to direct conversion, various methods are available that offer the advantage of reducing the bandwidth of the digitized signal. Omitting data that do not provide added value shortens the processing time required to analyze the raw data. The implementation of these methods will be discussed in detail later.

The third core component is a microcontroller, which ensures the previously specified process flow.

#### 2.3.3. Active Responder

Figure 6 shows the complete overview schematic of the active responder based on the heterodyne method. The signal path begins with an antenna, where the use of a planar antenna array is intended. A low-noise small-signal amplifier is placed before the first mixer for pre-amplification. The gain of this amplifier should not be too high, as mixers generally operate with less distortion at lower signal levels. The mixer’s role is to convert the signal to the intermediate frequency range. With sufficient separation between the high-frequency and intermediate-frequency bands, interference can be filtered more easily, but the power transferred into unwanted frequency ranges must be accounted for during design.

The local oscillator signal is generated using a phase-locked loop (PLL) to achieve a stable signal. Stability is desirable for both the phase and amplitude. Parameterization is performed via a microcontroller. The LO signal is coupled through an output of a power divider to the first mixer stage. Impedance transformation is necessary since the upconverting stage is also fed by the local oscillator, resulting in both mixers being connected in parallel from the LO’s perspective. Power leveling is performed in the intermediate frequency range, as control loops are generally easier to integrate at lower frequencies. This applies both to the expected availability of suitable components and to the extraction of the signal to be measured.

The control system typically consists of a variable, voltage-controlled attenuator, a subsequent amplifier stage, and a power detector that provides the control voltage for the attenuator. In this case, the control loop is fully analog.

As a result, the input signal for the subsequent second mixer stage, which performs the amplitude modulation and provides the desired additional information, is less subject to strong fluctuations. Even a free-running oscillator suffices as the identification oscillator since the distance information in FMCW measurements is determined from the difference between two points symmetrically distributed around $f_{\mathrm{AM}}$.

Since harmonic generation is expected, both from the IF input signal and the AM signal along with their cross-products, special attention must be paid to the operating point of this mixer. Narrowband filtering at the output removes unwanted components. The passband must accommodate the bandwidth of the RF signal, which in our case is 250 MHz. The stopband and cutoff are defined by the choice of the intermediate frequency band. Filtering reduces the bandwidth at the input of the subsequent amplifier stage, minimizing the consumption of unnecessary power reserves. This stage should be designed to maximize the drive level of the third mixer stage, minimizing the need for inefficient RF amplification. The transmitting antenna is DC-coupled to ground potential.

### 2.4. Block-Based Hardware Setup

A flexible approach has been adopted, allowing for modifications and measurements at various points in the signal path. This ensures adaptability and facilitates a deeper analysis of the system.

The radar system used was developed in-house, based on the previously introduced MMIC from Analog Devices. For high-frequency generation, the PLL formed with *ADF5901* and *ADF4159* was used, with *ADF5904* as the corresponding downconverter. Its basic design has been presented in [16]. This setup provides two TX channels and four RX channels, which are accessed via coaxial connectors, although only one channel is utilized for each. The board-to-connector transition is optimized according to [17,18]. This allows for the usage of various antenna types to handle radiation patterns, polarization as well as footprint size. Standard gain horn antennas and patch arrays as per [19] are intended for use. The remaining RF channels are disabled. The baseband is supplemented with instrumentation amplifiers, allowing the output signal from each downconverter to be accessed both differentially and single-ended. However, due to the differential input of the ADC used, the instrumentation amplifiers are not functional. The modular design results in the PCB stack shown in Figure 7, with the RF frontend on top, the ADC in the middle, and the microcontroller evaluation board mounted at the bottom on the carrier plate.

The ADC module contains the previously selected analog-to-digital converter of type *AD7264* in a single configuration, allowing a maximum of two baseband channels to be sampled. The stack order was chosen to minimize the trace lengths for the baseband and SPI bus between the microcontroller and ADC, as much as possible in a modular setup.

Figure 8 shows the RF module of the responder. The antennas are connected via the side-mounted coaxial couplers. This module performs RF amplification and mixing with the local oscillator signal, which is also generated on this board. At the intermediate frequency level, the downconverted signal is extracted, and the upconverted signal is injected on the backside via SMA coaxial connectors.

Figure 9 depicts the module carrier designed to accommodate up to four plug-in modules. The plug-in modules can be connected at all positions using SMA cables. Power is supplied via linear regulators located on the carrier board. In addition to a filter module to limit the bandwidth to the designated intermediate frequency range of 500–750 MHz, four additional components for signal amplification and attenuation were fabricated. The mixer used was tested with a plug-in LC oscillator.

## 3. Results

Two measurements were planned and carried out for the proof of concept. These should take into account the conceivable areas of application, so that the first setup was conducted in an outdoor environment as a more general measurement. This was followed by a further series of tests that took into account a specific area of application, namely by being carried out within a section of sewage pipe. This addresses the performance inside an area that is prone to clutter.

### 3.1. First Experiment: Free-Space Measurement

The first part of the experiment involved conducting free-space measurements. The goal of this measurement was to provide a general proof of functionality over distances ranging from approximately 10 to 100 m. Additionally, the relative accuracy of measurement points at various distances and the repeatability of the measurements were to be determined.

The corresponding block diagram is shown in Figure 10. According to this diagram, a radar base station with a PC for control and live evaluation, along with the active responder, supplemented by a spectrum analyzer, were set up on separate movable tables. The spectrum analyzer was fed by a signal directly tapped at the output of the receiving antenna via a 1:1 power splitter.

The specified distance was determined between the positions of the wheels of each table, which served as the reference plane. This approach was acceptable because the radar measurements inevitably exhibited a constant offset due to propagation delays caused by:The relative positions of the antenna reference planes;Cable connections between individual modules;The spatial dimensions of the electronics;Predefined zero points in the evaluation software.

All these factors remained constant during the experiment and resulted in a deviation that could be easily calibrated out.

The first measurement in the series was conducted at a distance of 1 m. Beyond 2 m, the distance was increased in 2 m steps. Additional measurements were performed at intermediate distances under special conditions. Since mains power was required at both tables, the experiment had to be interrupted at longer distances due to the limited length of power cables. Whenever an interruption occurred, the last measurement was repeated with unchanged component arrangements for verification.

The primary goal was to measure distances up to 100 m. However, due to the circumstances, additional measurements were conducted at distances up to 150 m.

At shorter distances, a characteristic signal shape in the frequency domain was observed. As expected, two prominent bins appeared sharply defined near the modulation frequency, tapering to noise towards the spectrum’s edges. For distances up to approximately 40 m, a characteristic “inverted bathtub curve” shape was visible in the spectrum. This curve, defined by steep gradients towards the symmetry axis and a curved drop-off to surrounding noise, contrasts with the typical bathtub curve where decaying edges are directed towards the center.

Beyond 41 m, a 2 m region of destructive interference was observed, causing the signal to disappear between 41 m and 43 m. Similarly, no evaluable signal was detected between 54 m and 62 m.

A cross-check of the spectrum of the baseband waveform showed that the signal flanks flattended for that certain range, forming a curved transition to noise at the inner edges, while the outer edges remained qualitatively unchanged. This made triggering to the correct threshold more difficult. Signal quality improved at greater distances, with the opposing edges of the spectral peaks becoming more steep and overall signal levels increasing. (Note that the modulation at the target shifts both range bins at the positve and negative frequencies by the modulation frequency, so the distance information is found in the relative position of these bins). A drop in signal quality was observed again between approximately 114 m and 118 m, roughly double the previously unevaluable range near 58 m, as shown in Figure 11.

In order to examine the system’s precision at small distance changes, the position of the responder unit was slightly varied in steps of approximately 1 cm. This was carried out in the setup of the first experiment, at distances of about 5 m and 10 m between the radar and the responder. The result is given in Figure 12, which indicates that the tolerance limits specified by the system requirements are exceeded in the test at 10 m.

### 3.2. Second Experiment: Measurement in PVC Pipes

The subsequent experiment investigated the applicability of the method inside water pipes. The available pipes are typically used to transport various types of wastewater, such as rainwater or sewage from residential areas. These pipes are generally buried underground during installation. For channel base pipes, the material PVC-U (unplasticized polyvinyl chloride) is frequently used. These extruded pipe sections have a multi-layer structure, with a foamed core surrounded by outer and inner layers, as per DIN EN 1401-1:2019-09 in the version valid at the time of the measurements [20].

The incident signal in the investigation of propagation within a pipe was tapped at the responder and observed using a spectrum analyzer. Additionally, the power in the baseband was determined based on the signal amplitude.

The raw data indicate a linear dependency of power in the logarithmic scale on distance in linear scaling. Consequently, it was decided to perform linearization immediately after logarithmizing the power. This relationship is clearly illustrated in the visualization of the measured data in Figure 13, which includes the regression.

As will be pointed out later in the discussion, this observation contradicts the Friis equation. In contrast to the result of the measurement, a distance dependency to the fourth power would normally be assumed for radar systems. Although the logarithmic decrease in power practically means stronger baseband signal amplitudes in the range of up to a few meters, it will drop significantly faster at greater distances.

## 4. Discussion

For the free-space test results, the measured carrier power values as well as distance-dependent power values determined at the base station were subjected to regression analysis. Since the power values were given on a logarithmic scale, the corresponding distance values were also logarithmized to base 10. In this double-logarithmic representation, the result was a straight line whose slope corresponded to the power of the distance. Thus, linear regression could be applied:(13)$$y∼\log_{10}\left(P\left(R\right)\right)∼-\log_{10}\left(R\right)$$

For power in dBm, the distance dependence can generally be expressed as:(14)$$P\left(R\right)=-m_{k}\cdot \log_{10}\left(\frac{R}{1\;m}\right)\cdot 10\;\mathrm{dBm}+b_{k}\cdot 1\;\mathrm{dB}$$

In free-space propagation, the coefficient $m_{k}$, which represents the distance exponent, is theoretically expected to be:(15)$$\begin{array}{cc} \mathrm{One}-\mathrm{way}\;\mathrm{propagation}:\; & m_{1}=2\\ \mathrm{Two}-\mathrm{way}\;\mathrm{propagation}:\; & m_{2}=4 \end{array}$$

The parameter $b_{k}$ depends on various factors, such as transmitted power, antenna gain, cable losses, and coupling to the measurement instrument. Its value plays a minor role and does not contribute to the evaluation of the transmission channel. The propagation type is indicated by the index *k*.

For the measured values, applying regression yields the following results for one-way and two-way propagation, denoted as $\tilde{P}$:(16)$$\begin{array}{cc} \mathrm{One}-\mathrm{way}\;\mathrm{propagation}:\; & m_{1}=1.585\;;\;b_{1}=-20.13\\ \mathrm{Two}-\mathrm{way}\;\mathrm{propagation}:\; & m_{2}=3.056\;;\;b_{2}=-22.35 \end{array}$$

Using logarithmic properties, the input power at the responder can be approximated as:(17)$${\tilde{P}}_{\mathrm{ARRS}}\left(R\right)=\log_{10}\left(\frac{1}{{\left(\frac{R}{1\;m}\right)}^{1.585}}\right)\cdot 10\;\mathrm{dBm}-20.13\;\mathrm{dB}$$

The output of the radar sensor is described as:(18)$${\tilde{P}}_{\mathrm{Base}}\left(R\right)=\log_{10}\left(\frac{1}{{\left(\frac{R}{1\;m}\right)}^{3.056}}\right)\cdot 10\;\mathrm{dBm}-22.35\;\mathrm{dB}$$

The drop in signal amplitude at certain ranges should be addressed as well. The second undetectable range is at about twice the distance of the first, so that a periodic behavior could be present. The next interval would be outside the selected measurement window. A likely explanation would be multipath propagation, which leads to destructive interference at certain distances. To confirm this, it would be conceivable to conduct another series of measurements in a different environment.

For the test series exectuted inside pipes, as before, the formal relationship between power in the logarithmic scale and a linear equation is established:(19)$$\begin{array}{cc} \tilde{P}\left(R\right)\cdot 1\;\mathrm{dBm} & =\log_{10}\left(\frac{P\left(R\right)}{1\;\mathrm{mW}}\right)\cdot 10\;\mathrm{dBm}\\ \; & =m_{k}\cdot \frac{R}{1\;m}\cdot 10\;\mathrm{dBm}+b_{k}\cdot 1\;\mathrm{dB} \end{array}$$

From the measured data at the responder and radar sensor, the following coefficients are determined:(20)$$\begin{array}{cc} \mathrm{One}-\mathrm{way}\;\mathrm{propagation}:\; & m_{1}=-0.09566\\ \mathrm{Two}-\mathrm{way}\;\mathrm{propagation}:\; & m_{2}=-0.1603 \end{array}\;\begin{array}{cc} \; & ;\;b_{1}=-18.04\\ \; & ;\;b_{2}=-21.57 \end{array}$$

Thus, after regression, the dependence of power $\tilde{P}$ in the linear representation on distance *R* as part of the exponent to base ten is expressed as follows for the input of the responder:(21)$$\begin{array}{cc} {\tilde{P}}_{\mathrm{ARRS}}\left(R\right) & =10^{-0.09566\cdot \frac{R}{1\;m}}\cdot 10^{-1.804}\cdot 1\;\mathrm{mW}\\ \; & =10^{-0.09566\cdot R/1\;m}\cdot 0.0157\;\mathrm{mW} \end{array}$$
and for two-way propagation as measured at the base station:(22)$$\begin{array}{cc} {\tilde{P}}_{\mathrm{Base}}\left(R\right) & =10^{-0.1603\cdot \frac{R}{1\;m}}\cdot 10^{-2.157}\cdot 1\;\mathrm{mW}\\ \; & =10^{-0.1603\cdot R/1\;m}\cdot 0.007\;\mathrm{mW} \end{array}$$

Based on the attenuation characteristics determined, it is reasonable to assume that there is a constant outflow of power per unit length from the inside of the pipe. The comparably poor high-frequency properties of polyvinyl chloride result in high attenuation within the wall. As the entire section is made up of individual plug-in elements, each 2 m long, there is also a discontinuity at each overlapping sleeve. The wall is interrupted there and the otherwise smooth inner surface is curled, which results in regular interference points. So far, it has not yet been investigated whether there is a propagation mode within the wall, which is quite possible given the operating frequency of the radar system and the wall thickness.

To focus on the accuracy of the sensor at small distance increments in the setup of the first experiment, the data analysis indicates deviations from the actual distance. The first data record was acquired at about 5 m, the deviations from the actual distance reached up to 3 cm, which is acceptable with regard to the system requirements. However, at a range of 10 m, deviations increased, with some readings differing by as much as 7 cm. Despite this, the goal was to achieve a deviation of less than 5 cm, which was not fully met in this case. The results highlight the challenges of achieving the accuracy required with the given signal processing. Since the focus of this work was expressly not in the data-processing domain, its optimization should be clearly addressed in future works. Provided the frequency band and bandwidth are not to be changed, this will be a necessary step to achieve a significant improvement of absolute tolerances.

## 5. Conclusions

In this work, the *Active Radar Interferometer (AcRaIn)* was presented as an innovative method in secondary radar technology. Overall, a compact design is possible, as the peripheral components around the technological core are minimized: the processing of supply voltages and the one-time parameterization of the integrated control circuits during startup.

As preparation for the experimental setup, several preliminary considerations were made. After defining the technical requirements, a block diagram was derived that takes advantage of the heterodyne method. The operating frequency of the radar was chosen to be in the ISM band at $f_{c},\mathrm{HF}=24.125\;\mathrm{GHz}$ with a bandwidth of BW=250MHz.

The functionality of the entire system was subsequently confirmed in various measurement series. Several scenarios were tested in free space and in various above-ground pipe sections made from assembled plastic pipe pieces. In free space, an outstanding result was achieved, with a clear signal being evaluated even at a distance of 150m between the radar sensor and responder. In another series, measurements were conducted with respect to a potential application area, specifically the use within a water pipeline, where the distance to a reference point at the access shaft was to be measured.

As a result of these investigations, in addition to the implementation of efficient signal processing, the effect of logarithmic signal attenuation should be questioned and explained from the perspective of wave theory.

## Figures and Tables

**Figure 1 sensors-25-01711-f001:**
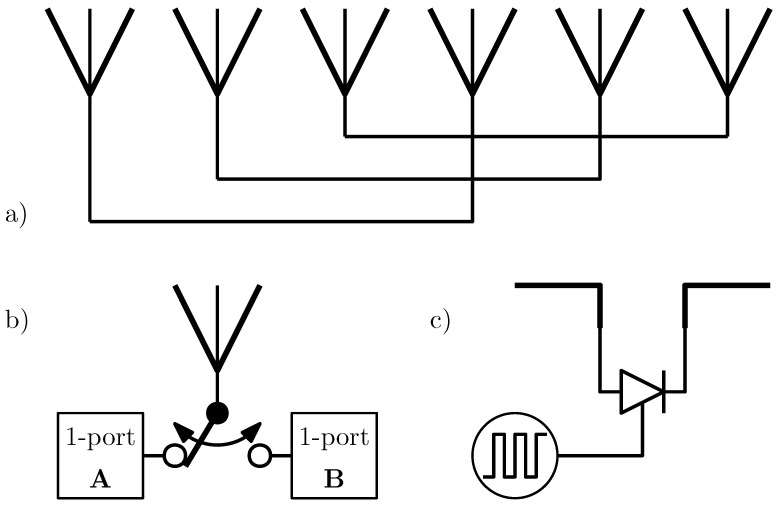
Examples of possible arrangements for passive and semi-passive backscatterers: (**a**) Van Atta array, (**b**) variation of antenna termination, (**c**) switching diode as load.

**Figure 2 sensors-25-01711-f002:**
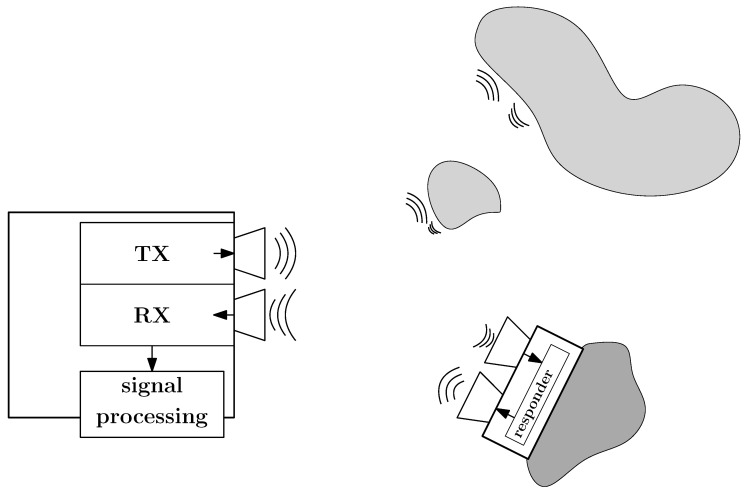
Block diagram of the system.

**Figure 3 sensors-25-01711-f003:**
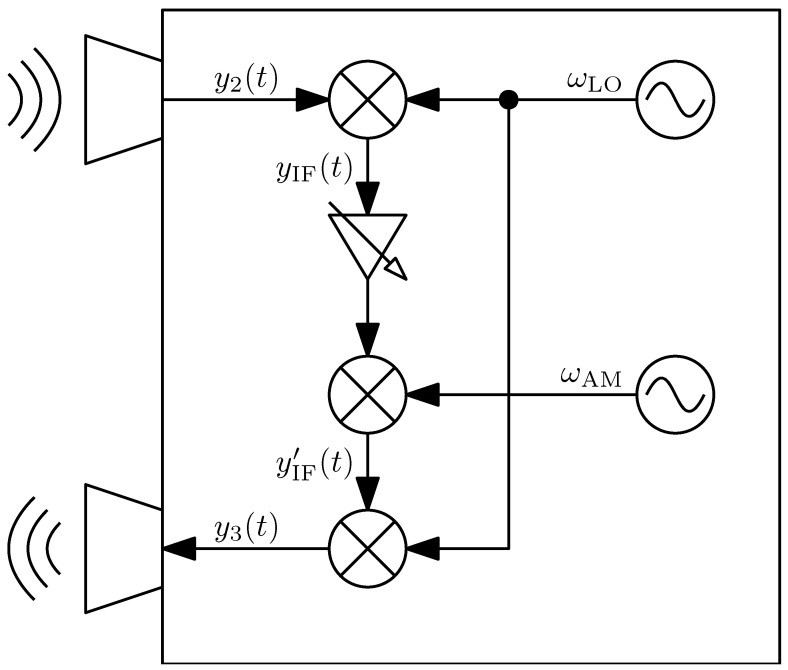
Active radar responder with amplitude modulation based on the heterodyne principle to achieve unambiguity.

**Figure 4 sensors-25-01711-f004:**
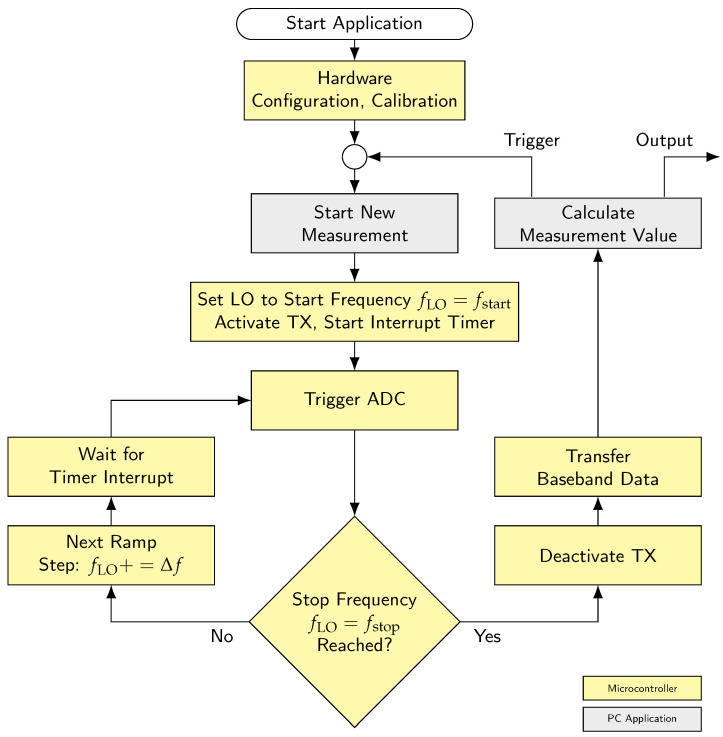
Workflow and interaction between PC application and base station.

**Figure 5 sensors-25-01711-f005:**
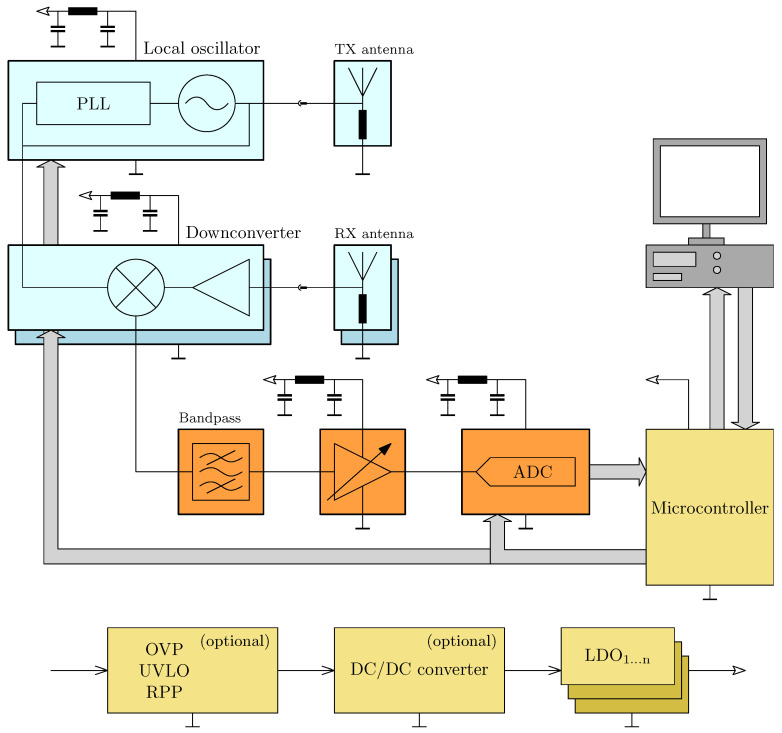
Block diagram of the radar frontend.

**Figure 6 sensors-25-01711-f006:**
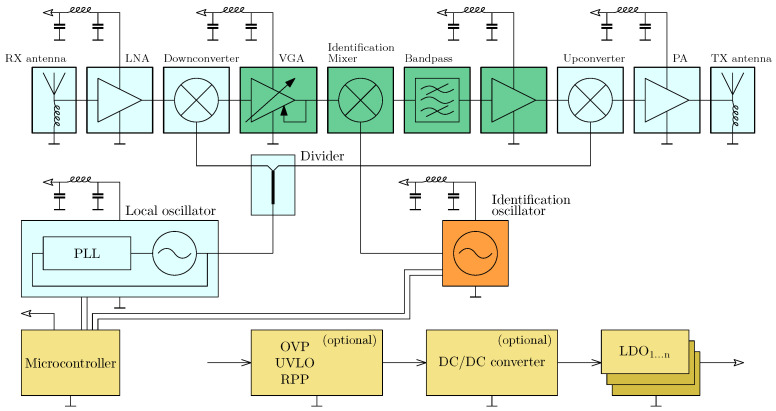
Complete block diagram of the active radar responder.

**Figure 7 sensors-25-01711-f007:**
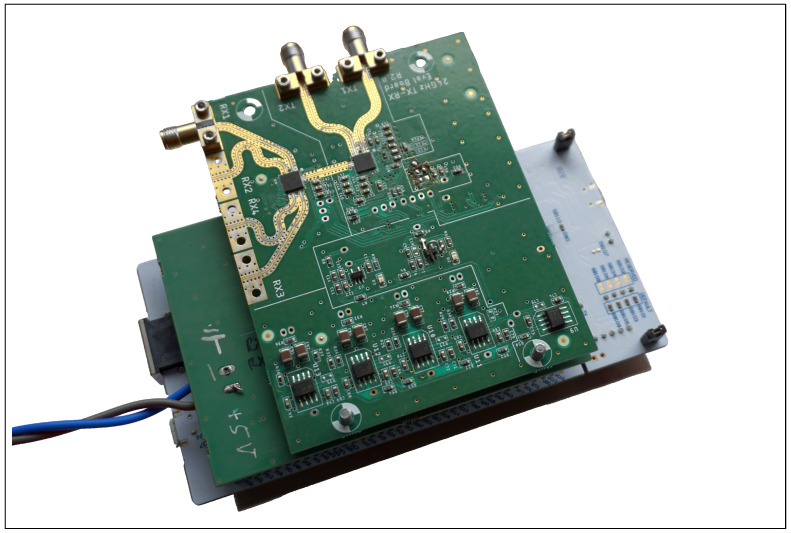
Fully assembled radar base station with RF frontend on ADC and microcontroller boards.

**Figure 8 sensors-25-01711-f008:**
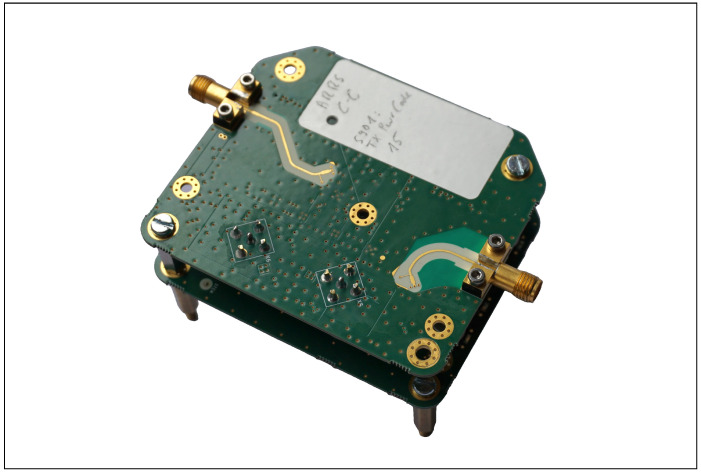
RF module of the responder, with the secondary board containing the switching power supply and protection circuitry below.

**Figure 9 sensors-25-01711-f009:**
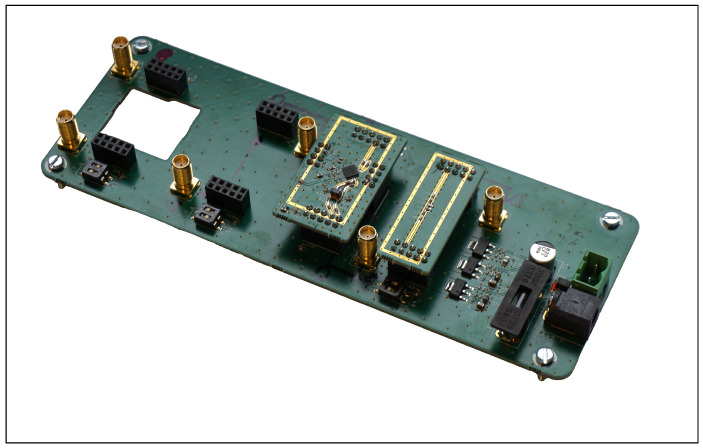
Intermediate frequency carrier module of the active responder for up to four plug-in modules, shown here with mixer and bandpass filter installed.

**Figure 10 sensors-25-01711-f010:**
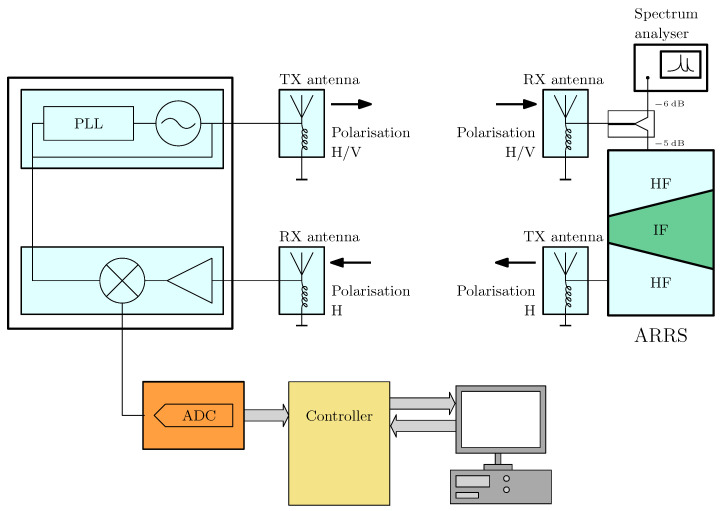
Block diagram of the experimental setup.

**Figure 11 sensors-25-01711-f011:**
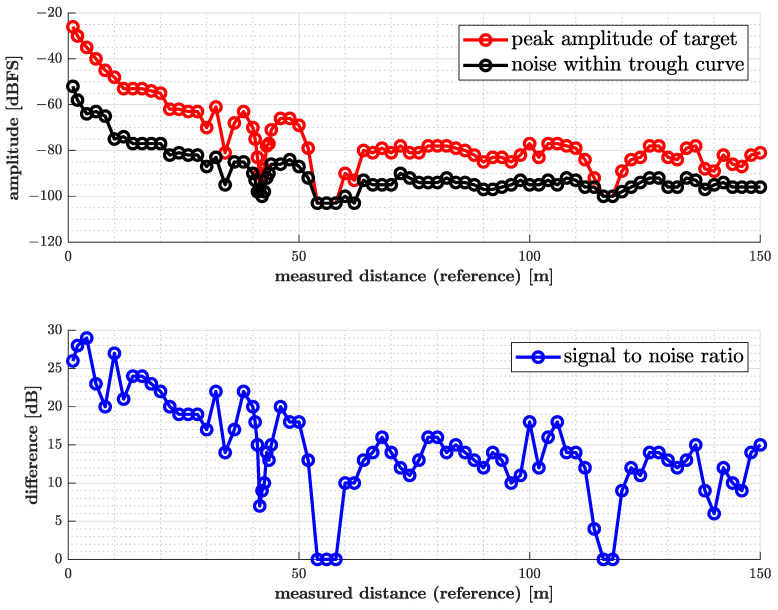
Summary of results from free-space measurements with orthogonal antenna polarization.

**Figure 12 sensors-25-01711-f012:**
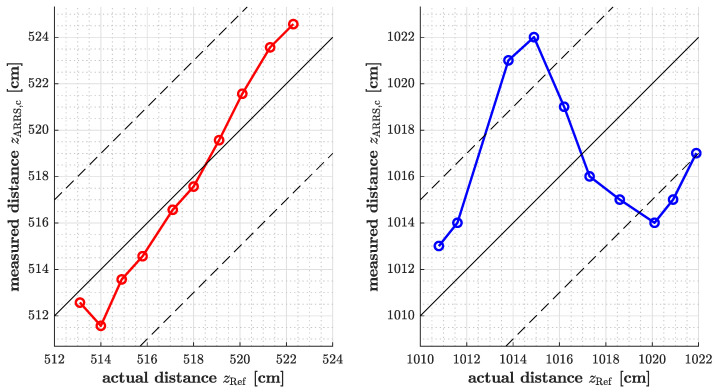
Measured vs. actual distance for small-range variations.

**Figure 13 sensors-25-01711-f013:**
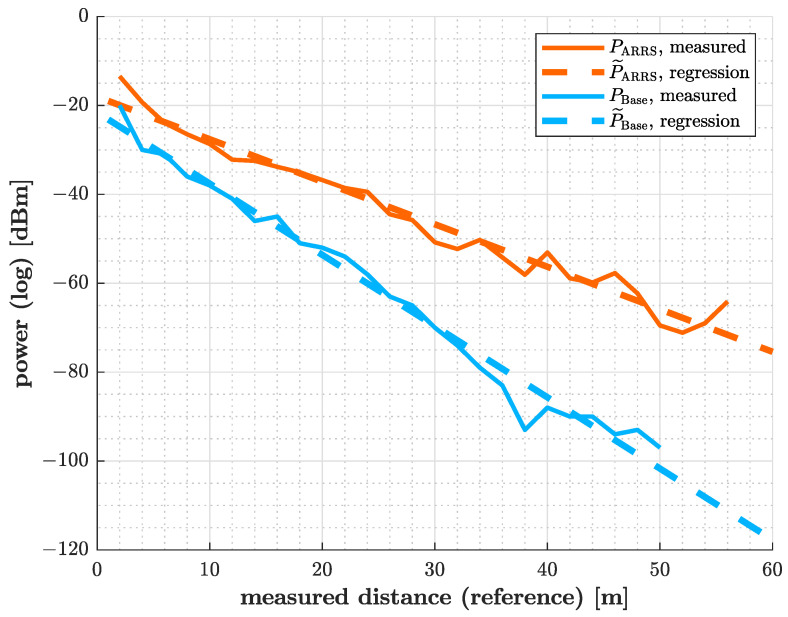
Regression of power during one-way and two-way propagation in the plastic pipe.

**Table 1 sensors-25-01711-t001:** List of variables.

A,B,D	amplitudes of high-frequency signals
$b_{k},m_{k}$	linear equation coefficients
BW	bandwidth of an FMCW ramp
$c_{0}$	speed of light
$f_{l}$, $f_{u}$	frequency of left-hand and right-hand terms in baseband
*P*	power
$\tilde{P}$	regression of measured power values
*R*	range
$T_{\mathrm{oF}}$	time of flight (one direction)
$\Delta T_{c}$	duration of one FMCW ramp
$u_{l}\left(t\right)$, $u_{u}\left(t\right)$	left-hand and right-hand terms in baseband
$y_{1}\left(t\right)$	TX signal of the radar sensor
$y_{2}\left(t\right)$	incident signal at the radar target
$y_{1}\left(t\right)$	TX signal of the radar target
$y_{4}\left(t\right)$	incident signal at the radar sensor
$y_{\mathrm{IF}}\left(t\right)$	downconverted intermediate signal at the radar responder
$y_{\mathrm{IF}}^{\left[\mathrm{AM}\right]}\left(t\right)$	AM modulated intermediate signal at the radar responder
$\omega_{\mathrm{AM}}$	identification frequency of the radar responder
$\omega_{\mathrm{HF}}$	high frequency oscillation of the radar sensor
$\omega_{\mathrm{LO}}$	local oscillator frequency of the radar responder
φ	phase of an oscillation
ΔΦ	phase modulation caused by doppler shift

## Data Availability

Any research data not explicitly referenced or included in this publication is subject to a nondisclosure agreement and, as such, cannot be published or shared without prior authorization. All data covered under this agreement remains confidential and is protected from public dissemination.

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
