# Peer review of "An Active Radar Interferometer Utilizing a Heterodyne Principle-Based Target Modulator"

_sensors, 2025, doi:10.3390/s25061711_

Round 1

Reviewer 1 Report

Comments and Suggestions for Authors

Dear Authors,

I carefully read your paper and have a few minor comments. 

Line 36: An citation to the original work of Van Atta would be appropriate here.

Line 47: \Lambda is not defined? Usually, lowercase \lambda is used for wavelength

Line 59: Also here, some references (Tachymeter etc.) would be appropriate I would suggest

Line 131:  equation should have a lowercase "e" here...

Line 320: The heading "flexible approach" is not specific enough/unclear in my opinion

Line 354: The meaning of the sentence "This section may be..." is completely unclear to me

Line 366: ...analyzer was fed BY a signal... ("by is missing")

Section 3.1: A few sentences/comparison to the well known Friis equation and Radar equation would be very helpful

Otherwise, the paper is well written and organized. It is also a lot of work necessary to setup such a system and get it working in a real-world environment. This renders the paper valuable.

Best regards

Reviewer 2 Report

Comments and Suggestions for Authors

In this paper, the active radar interferometer is presented as an innovative method in secondary radar technology, and the functionality of the system was confirmed in various measurement series. However, there are many issues need to be addressed in its present form. The main comments are listed as follows.

  1. In the introduction, only the preceding development is given, but the main innovative work of this paper is not clearly explained.
  2. The Equations appear abrupt and lacks corresponding derivation or explanation, and there is a lack of explanation for the corresponding symbols, such as ToF, ωHF,
  3. There are shortcomings in the discussion and analysis of the experimental results. For example, why a drop in signal quality between 41m and 43m, 114m and 118 m? How to understand “Signal quality improved at greater distances ….”, “…… roughly double the previously unevaluable range near 58 m.”
  4. The description between lines 443 and 452 is confusing, and the main conclusion drawn from the experiment is not very clear.
  5. On line 188, how to understand “the resolution is ±5cm”?
Comments on the Quality of English Language

The English could be improved to more clearly express the research.

Round 2

Reviewer 2 Report

Comments and Suggestions for Authors

All my comments have been considered by the authors, and the response is satisfactory.